# Necroptosis Blockade Potentiates the Neuroprotective Effect of Hypothermia in Neonatal Hypoxic-Ischemic Encephalopathy

**DOI:** 10.3390/biomedicines10112913

**Published:** 2022-11-13

**Authors:** Mathilde Chevin, Stéphane Chabrier, Marie-Julie Allard, Guillaume Sébire

**Affiliations:** 1Department of Pediatrics, Research Institute of the McGill University Health Centre, McGill University, Montreal, QC H4A 3J1, Canada; 2CHU Saint-Étienne, INSERM, Centre National de Référence de l’AVC de l’enfant, CIC1408, F-42055 Saint-Étienne, France; 3INSERM, Université Saint-Étienne, Université Lyon, UMR1059 Sainbiose, F-42023 Saint-Étienne, France

**Keywords:** HI, HT, necroptosis, Nec-1, MLKL, RIPK, TNF, neuronal death, neuroinflammation, neonatal encephalopathy, therapeutics, brain injury, neurodisability, neuroprotection

## Abstract

Neonatal encephalopathy (NE) caused by hypoxia-ischemia (HI) affects around 1 per 1000 term newborns and is the leading cause of acquired brain injury and neurodisability. Despite the use of hypothermia (HT) as a standard of care, the incidence of NE and its devastating outcomes remains a major issue. Ongoing research surrounding add-on neuroprotective strategies against NE is important as HT effects are limited, leaving 50% of treated patients with neurological sequelae. Little is known about the interaction between necroptotic blockade and HT in neonatal HI. Using a preclinical Lewis rat model of term human NE induced by HI, we showed a neuroprotective effect of Necrostatin-1 (Nec-1: a compound blocking necroptosis) in combination with HT. The beneficial effect of Nec-1 added to HT against NE injuries was observed at the mechanistic level on both pMLKL and TNF-α, and at the anatomical level on brain volume loss visualized by magnetic resonance imaging (MRI). HT alone showed no effect on activated necroptotic effectors and did not preserve the brain MRI volume. This study opens new avenues of research to understand better the specific cell death mechanisms of brain injuries as well as the potential use of new therapeutics targeting the necroptosis pathway.

## 1. Introduction

Hypoxic-ischemic encephalopathy (HIE) is a subset of neonatal encephalopathy (NE) resulting from hypoxic-ischemic (HI) injuries of term newborns [1,2]. HIE is characterized by severe neurological symptoms at birth and poor long-term outcomes, including cerebral palsy, learning disabilities, and behavioural impairments [3,4]. NE affects up to 1% of live births, and HIE accounts for about 20% of NE cases [1,2,4,5]. Treatment options of HIE are mostly symptomatic. The only curative therapeutic option consists of therapeutic hypothermia (HT), which is now a mandatory standard of care [6,7,8,9]. However, HT provides only limited neuroprotection. Around 50% of HT-treated neonates will experience moderate to severe neurologic disabilities [6,7,8,10].

Necroptosis refers to a recently discovered pathway of programmed cell death, which seems to play a role in various neurologic conditions including amyotrophic lateral sclerosis (ALS), Alzheimer disease (AD), Parkinson disease (PD), and multiple sclerosis (MS), as well as acute inflammatory and neurological conditions such as traumatic brain injuries and stroke [11,12,13,14,15,16,17,18]. It differs from apoptosis by its earlier onset (<6 h post-injury) and the involvement of a specific mechanistic cascade [19,20,21,22]. It is still unknown if necroptosis could play a role in term newborns suffering from HIE. However, such early cell death mechanisms could be at the origin of HIE-induced neonatal brain injuries [8,23,24,25,26]. This hypothesis is supported by preclinical data showing that necroptosis is implicated in some acute adult brain injuries, including cerebral ischemia and neurodegenerative diseases (such as MS, AD, and PD) [13,18,27,28,29,30], as well as some evidence from preterm HI-induced injuries [24,31,32,33]. Key necroptotic markers, including receptor-interacting protein kinases (RIPK)1/3, and mixed lineage kinase domain-like protein (MLKL), have been found to play an important role in human neurodegenerative disease progression and neural cell death [18,34].

Altogether, these findings prompted us to focus our research on the unexplored role of necroptosis in HIE of term newborns using our well-established rat model of HIE [35,36,37,38,39].

To test the involvement of necroptosis in HIE, we took advantage of the necroptotic blocking agent Necrostatin-1 (Nec-1) to modulate the necroptotic cascade [13,31,32,40,41,42]. Due to the dual role of RIPK1 on inflammation and of driving cell death, Nec-1 can also reduce cerebral apoptosis, oxidative stress, and inflammation in adult models of stroke and HI injuries [13,20]. Using a multimodal approach (*i.e.*, histology, immunohistochemistry [IHC] and small animal imaging), we assessed the efficacy of Nec-1 in combination, or not, to HT. Beyond understanding the mechanisms leading to brain injury, the study of necroptosis in HIE might open new neuroprotective strategies to alleviate HIE refractory to HT treatment only.

## 2. Materials and Methods

The datasets used and analyzed during the current study are available from the corresponding author on reasonable request.

### 2.1. Rat Model

The experimental protocol was approved by the Institutional Animal Care Committee of McGill University (protocol #2015-7691) in accordance with the Canadian Council on Animal Care Guidelines (accessed on 12 November 2022): https://ccac.ca/en/standards/guidelines/.

A total of 88 Lewis rat pups at postnatal day (P)6 from 13 different litters, were obtained from Charles River Laboratories (Kingston, NY, USA). At P12, corresponding to the level of rat brain development equivalent to that of a term human newborn [43,44], HI was induced by permanent ligation of the right common carotid artery (RCCA) followed by 8% O_2_ exposure at 36 °C for 1.5 h (Figure 1a) [35,36,37,38,39,45]. Nec-1 was purchased from Sigma-Aldrich (Oakville, ON, Canada) and was dissolved in DMSO. Before intraperitoneal injection, it was diluted in pyrogen-free saline to reach a final concentration of 10 mg/kg, as previously used in adult model of subarachnoid hemorrhage [46]. Considering the very short half-life of Nec-1 (90 min post-injection), we administrated Nec-1 twice, before and after hypoxia induction [47,48] (Figure 1a).

A control (CTL) group received an equal volume of DMSO and saline, and underwent a sham surgery, consisting of the right common carotid artery exposure (i.e., without ligation, hypoxia, and HT) (Figure 1a). HT was induced 30 min after hypoxia, as previously described [35,37,39,45]. Pups were kept on a hot plate at 28 °C for four hours to lower their core body temperature to 32–33 °C (Figure 1b) [16,35,37,39,49]. There was no mortality following surgery. Two rats (one HI+HT, one HI+Nec-1+HT) died during imaging acquisitions and were removed from the study. Their death occurred eight days following HI exposures and was not considered as related to treatment. All experimenters were blind to experimental conditions during data analysis.

### 2.2. Histology

The brains were removed and fixed (paraformaldehyde 4%, glutaraldehyde 0.1%) at room temperature, paraffin-embedded, and sectioned in 5-μm thick slices using a microtome, as described [35,45]. IHC staining was performed to visualize necroptotic markers at 5 h post-HI, in six to eight per experimental conditions (CTL, HI, HI+HT, HI+Nec-1, and HI+Nec-1+HT) from eight different litters. Coronal sections were scanned at the epicentre of the infarct (−2.30 mm to −2.50 mm from the Bregma) using a NanoZoomer Digital Pathology Scanner (Hamamatsu Photonics, Japan), as described [35,45].

### 2.3. Immunohistochemistry

IHC was performed as described [16,35,36]. Briefly, sections were incubated for 2 h at room temperature with the antibodies (Appendix A). The appropriate horseradish peroxidase-conjugated secondary antibodies (Appendix A) were used for a 45 min incubation at room temperature. IHC labelling was revealed using diaminobenzidine (DAB) (Roche, QC, Canada). Slides were counterstained with hematoxylin. Four fields from the parietal cortex and the caudate-putamen and two fields from the hippocampus were counted (−2.30 mm to −2.50 mm from the Bregma). The expression of the proteins of interest (i.e., tumor necrosis factor [TNF]-α, phosphorylated [p]RIPK3, and pMLKL) was analyzed using the Image J analysis software (NIH Image, https://imagej.nih.gov/ij/ (accessed on 5 March 2020), as previously described [16,35,36,50]. Briefly, quantitative comparisons were performed between high-positive staining intensities from HI±Nec-1±HT conditions using the IHC profiler tool [51]. Measurements were corrected based on the mean expression of proteins of interest in the CTL group and expressed as percentages.

### 2.4. Magnetic Resonance Imaging (MRI) of the Rat Brain

MRI scans were performed at P20-21 using the 7T Bruker BioSpec 70/30USR system (Bruker Biospin, Ettlingen, Germany) at the Research Institute of the McGill University Health Centre Small Animal Imaging Labs (http://rimuhc.ca/small-animal-imaging-labs) (accessed on 5 March 2020), as previously described [39]. Briefly, MRI studies were performed under 1.8–2% isoflurane with the respiration rate controlled at 55–60 breaths/min and the body temperature maintained at 37 ± 0.3 °C. Anatomical images were acquired using a 3D balanced Steady-State free precession (bSSFP) sequence with repetition time = 5.2 mm, echo time = 2.6 mm, flip angle = 30*, matrix size = 192 × 192 × 192 mm; NEX = 4, field of view = 44.41 × 44.41 × 22.20 mm; spatial resolution = 231 × 231 × 116 µm; and acquisition time 28 min.

### 2.5. Analysis of MRI Data

Image analysis was performed using the SPM8 software package (https://www.fil.ion.ucl.ac.uk/spm) (accessed on 5 March 2020), and an image analysis toolbox developed in house based on Mathworks’ MATLAB (https://www.mathworks.com) (accessed on 5 March 2020). Medical imaging software Dragonfly version 2022.1 (https://www.theobjects.com/dragonfly/index.html (accessed on 5 March 2020), Montreal, QC, Canada) and MRIcron (v1.0.20190902; http://www.mccauslandcenter.sc.edu/mricro/ (accessed on 5 March 2020)) were also used to generate illustrations. All DICOM images from MRI scanner were converted into NIFTY format to conduct image analysis.

#### 2.5.1. Measurement of the Stroke Volume

The stroke volume was measured using MR anatomic images at P20-21, i.e., eight days after exposure to HI. T2 hyperintensity was used as an effective marker to identify the boundaries of the lesion area on MRI, which were easily segmented using our image processing toolbox, as previously described [39].

#### 2.5.2. Region-of-Interest (ROI) Analysis of MRI Images

ROIs were manually traced using Dragonfly segmentation tools. Regional volume measurements were extracted for whole hemispheres, hippocampus and caudate-putamen area based on MRI/DTI Atlas of the rat brain (Paxinos et al., 2015) from Bregma 3.39 to Bregma –8.36. Right versus left areas were compared using the following ratio:(1)$$\left(\frac{\left(\mathrm{Left}\;\mathrm{area}-\mathrm{Right}\;\mathrm{area}\right)}{\mathrm{Left}\;\mathrm{area}}\right)\times 100$$

### 2.6. Data Analysis

Statistical analyses were performed using IBM Statistics 27 (SPSS) and GraphPad software version 9. The normality of residuals was tested by Shapiro–Wilk tests. Male and female data were pooled, as no significant interaction was detected between sex and treatment. The sex ratio was similar between experimental conditions. Outliers were removed from the data analysis by performing Grubbs’ test. Data were analyzed by one-way analysis of variances (ANOVA) or the Kruskall–Wallis test when normality assumptions were not met. When significant, pairwise comparisons were performed using Tukey’s HSD or Dunn’s tests. Data were presented as the mean ± standard error of the mean (SEM). Statistical significance level was set at *p* ≤ 0.05.

## 3. Results

### 3.1. Necroptotic Markers Measurements in HI±Nec-1±HT

Key necroptotic markers, i.e., TNF-α, pRIPK3 and pMLKL, were stained in the right brain hemisphere exposed to HI. Targeted ROIs were the somatosensory cortex, caudate-putamen and hippocampus.

#### 3.1.1. TNF-α Measurement in HI±Nec-1±HT

After HI exposure, the TNF-α expression was upregulated within the somatosensory cortex, caudate-putamen, and hippocampus (Figure 2a). Nec-1+HT treatment decreased TNF-α staining within the hippocampus, compared to HT treated rat pups (*p* < 0.05; Figure 2a). There was a trend toward a decreased expression of TNF-α within the somatosensory cortex between HI+Nec-1+HT versus HI+Nec-1 conditions (*p* = 0.06).

#### 3.1.2. pRIPK3 Measurement in HI±Nec-1±HT

After HI-exposure, pRIPK3 staining was upregulated within the selected ROIs. After HI exposure, a trend (*p* = 0.09) toward a decreased expression of pRIPK3 was detected within the somatosensory cortex of Nec-1+HT-treated rat pups (data not shown).

#### 3.1.3. pMLKL Measurement in HI±Nec-1±HT

After HI exposure, the expression of pMLKL (i.e., the key effector of necroptosis) was increased within the somatosensory cortex, caudate-putamen and hippocampus (Figure 3a). HT alone was not efficient for downregulating pMLKL expression (Figure 3b) in HI-exposed pups. However, Nec-1-treated pups had significantly reduced pMLKL expression within the somatosensory cortex, caudate-putamen, and hippocampus (Figure 3b). As compared to sole HT, Nec-1+HT-treated pups had significantly reduced pMLKL expression within the somatosensory cortex, and a trend (*p* = 0.06) toward reduced expression in the caudate-putamen (Figure 3b). pMLKL staining seems to be mainly neuronal, as described recently in a rat model of stroke [30].

### 3.2. Effect of Nec-1+HT on HI-Induced Stroke

The ratio of the right and left volume measurements, extracted for the whole hemispheres, caudate-putamen, and hippocampus, was performed at P20 (Figure 4a). Following HI exposure, Nec-1+HT-treated rats had significantly reduced right hemisphere, caudate-putamen, and hippocampus tissue loss (Figure 4b, *p* ≤ 0.05). Compared to HT alone, double-treated Nec-1+HT rats had less right caudate-putamen and hippocampal tissue loss (Figure 4b, *p* ≤ 0.05). As compared to HI, Nec-1-treaded pups showed a trend (*p* = 0.06) toward reduced tissue loss in the caudate-putamen.

## 4. Discussion

Using a preclinical rat model of HIE, we showed a neuroprotective effect of Nec-1 in combination with HT. This double treatment reversed the activation of key necroptotic markers and prevented the atrophy of the right hemisphere, caudate-putamen, and hippocampus of HI-exposed rats. HT alone did not show any effect on activated necroptotic effectors and did not impact brain volume.

The beneficial effect of Nec-1 added to HT against HIE was observed on both pMLKL and TNF-α at 5 h post-HI and MRI volumes at P20-21, within the same spatial distribution (i.e., somatosensory cortex, caudate-putamen, and hippocampus). To our knowledge, this is the first preclinical study demonstrating such potentialization of combined HT+Nec-1. In addition, we previously observed that HT alone might not impact the necroptosis pathway, and early neuroinflammation [35].

Such promising evidence is crucial to further investigate the potential synergic effect of necroptotic inhibitors on HT [8,24,25,26]. Two preclinical studies demonstrated a neuroprotective effect of Nec-1 in mice at P7 on oxidative damage, mitochondrial dysfunction, inflammation, and brain injury following neonatal HI [31,32]. In contrast to these studies, we did not observe significant Nec-1 effects on brain injury via MRI volume. However, these studies used a different experimental design. They focused on preterm mice pups at P7 exposed to a single intracerebroventricular injection of Nec-1 after 45 min of hypoxia. Furthermore, measuring surfaces (but not volumes) of a few selected forebrain sections stained with cresyl violet induces a potential selection bias in the data analyses. Using a term rat HIE model, Qu and colleagues recently showed that the inhibition of MLKL using siRNA was able to attenuated the infarcted volume using triphenyltetrazolium chloride staining, 7 days after HI [33].

Even if HT is already used as the standard of care in human neonatal HIE, recent preclinical evidence suggests a more modest neuroprotective effect of HT in HI than initially expected [52]. In contrast with the few studies dealing with the effect of HT alone in HI-exposed pups, we did not find sex-specific differences in the parameters we assessed [52,53,54]. However, the age of pups (P7 versus P12) and differences in the experimental HT procedure could explain these different outcomes related to sex [43,52,53,54].

### Limitations and Future Directions

While presenting innovative and promising findings, our study only focused on necroptosis and key activated necroptotic markers at 5 h post-HI. However, cell death after neonatal HIE represents a continuum of different mechanisms occurring at specific time points and involving several types of cell [8,24,25,26]. For instance, neuroinflammation, apoptosis and necroptosis are very intricated together, leading to significant brain injuries following acute injury as seen after HIE (Figure 5) [8,25,26]. Additionally, our study involved in situ IHC methods, but mRNA explorations were not carried out. Future studies should ideally consider testing for gene expressions involved in necroptosis-induced brain injuries in HIE.

Among several necroptotic inhibitors, we decided to use the RIPK1 inhibitor Nec-1 as it was shown to block necroptosis efficiently in vivo in some neurological and extra-neurological conditions [11,13,41,42,55]. Due to the dual role of RIPK1 on inflammation and cell death, Nec-1 is also able to reduce cerebral apoptosis, oxidative stress, and inflammation in adult models of stroke and HI injuries [13,20,31,32,56]. In addition, NEC-1 (i) blocked oligodendrocyte death, microglial inflammation and axonal degeneration in ALS models [16,57,58], (ii) protected against oligodendrocyte death in MS models [59], (iii) reduced the β amyloid burden, levels of inflammatory cytokines, and memory deficits in an AD model [60], and (iv) attenuated MPTP-induced dopaminergic neuronal loss in a PD model [61]. As compared to RIPK3 and MLKL inhibitors, RIPK-1 inhibitors such as Nec-1 downregulate apoptosis [21,32] are effective on rodents [41] and emerge as a promising therapeutic compound for the treatment of a wide range of human diseases [18,42,62].

HT alone might have neuroprotective effects on apoptosis [6,7,8], on reduction of oxidative stress through the increase of antioxidant enzymes [35,63], and on brain metabolic activity [39,64,65]. In the present study, we did not find any evidence of a neuroprotective effect of HT or Nec-1 alone but did observe a potentialization of HT combined with Nec-1. We cannot exclude a differential impact of HT on oxidative stress and apoptosis, as well as Nec-1 on oxidative stress, neuroinflammation and necroptosis (Figure 5). Both treatments could also affect delayed cell death occurring through both apoptosis and necroptosis [8,25,26].

**Figure 5 biomedicines-10-02913-f005:**
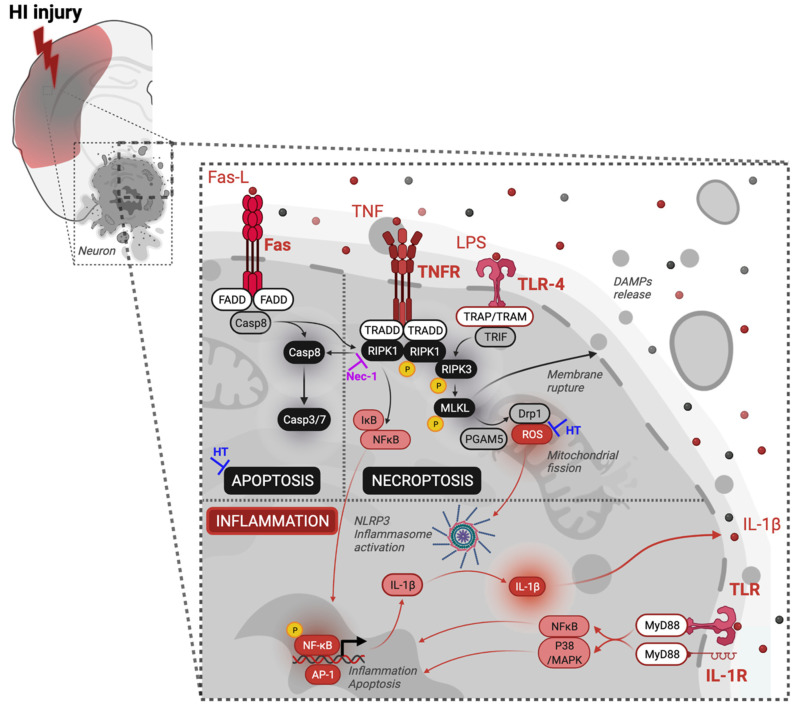
Schematic hypothesis of the continuum necroptosis/apoptosis/inflammation of neuronal cell death in HIE. Necroptosis is triggered by inflammatory mediators, including Fas-L and TNF-α. Upon the recruitment of adapter proteins by the ligands mentioned above, it will induce the phosphorylation of RIPK1. RIPK1 is a pivotal kinase that could induce (i) inflammation via the activation of NFκB [34], (ii) apoptosis when caspase-8 is activated, or (iii) necrosome formation if caspase-8 is inhibited. The necrosome is composed of interaction and activation of RIPK1 and RIPK3, followed by the recruitment and phosphorylation of MLKL. The necrosome will then induce necroptosis of the cell through upregulation of ROS and/or mitochondrial fission through interaction with mitochondrial phosphatase PGAM5 and Drp1 activation. Phosphorylated MLKL can also initiate cell death by disrupting the plasma membrane integrity [21,30,66,67,68,69,70,71]. Abbreviations: Drp1: dynamin-related protein 1; FADD: Fas-Associated protein with Death Domain; Fas-L: Fas-ligand; MLKL: mixed-lineage kinases domain-like; Nec-1: necrostatin-1; NFκB: nuclear factor κB; p: phosphorylated; PGAM5: phosphoglycerate mutase family member 5; RIPK: receptor-interacting protein kinases; ROS: reactive oxygen species; TNF-α: tumour necrosis factor α; TNFR1: tumour necrosis factor receptor 1; TRADD: Tumour necrosis factor receptor type 1-associated DEATH. Created with BioRender.com (access on 10 May 2021).

## 5. Conclusions

Our results showed for the first time the Nec-1 potentiation of hypothermia activity against neonatal HIE. The beneficial effect of Nec-1 added to HT against NE injuries was observed at the mechanistic (decreased of key neuroinflammatory markers) and anatomical (reduction in brain volume loss on MRI) levels. HT alone showed no effect on activated necroptotic effectors or MRI brain volume preservation. This study opens new avenues of research to better understand these mechanisms of cell death after acute neonatal brain injury. RIPK1 inhibitors are currently being investigated in several human clinical trials for neurodegenerative diseases, such as ALS and AD; and autoimmune diseases, such as psoriasis, rheumatoid arthritis, and ulcerative colitis (Clinical trials: NCT02776033, NCT02858492, NCT02903966, NCT03757351, and NCT03757325). Additional research will be crucial to investigate the efficacy and safety of RIPK1 inhibitors used as monotherapy or as combination therapy in various immunological and CNS diseases.

## Figures and Tables

**Figure 1 biomedicines-10-02913-f001:**
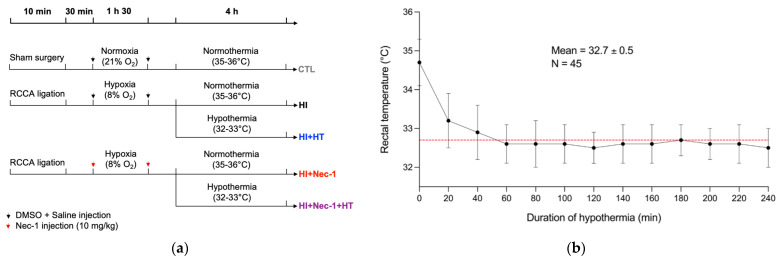
Experimental design. (**a**) Ligature of the right common carotid artery (RCCA) was performed on Lewis rat pups at P12. Necrostatin-1 or saline was injected intraperitoneally (i.p.) before and after hypoxia (8% O_2_ for 1.5 h). Rat pups were treated or not by hypothermia for 4 h. For control animals, a sham surgery was followed by i.p. saline injections, without hypoxia nor hypothermia. (**b**) Rectal temperature of every pup that underwent hypothermia (n = 45 from 11 litters). The mean temperature was 32.7 °C ± 0.5 °C. Abbreviations: CTL: control; HI: hypoxia-ischemia, HT: hypothermia; Nec-1: necrostatin-1; RCCA: right common carotid artery.

**Figure 2 biomedicines-10-02913-f002:**
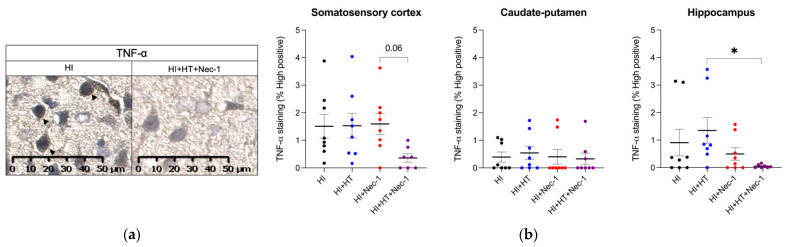
TNF-α expression in the brain at 5 h post-HI from pups exposed to HI±Nec-1±HT. (**a**) Representative image of a TNF-α staining in the somatosensory cortex of HI compared to HI+HT+Nec-1-exposed animals (**b**) Percentage of TNF-α staining compared to controls within the somatosensory cortex, caudate-putamen, and hippocampus in HI±Nec-1±HT conditions. The number (n) of rats used was n = 7–8 per conditions from 8 litters. The bars indicate the mean ± SEM. * *p* ≤ 0.05, Kruskal–Wallis with Dunn’s multiple comparisons tests. Abbreviations: CTL: control; HI: hypoxia-ischemia, HT: hypothermia; Nec-1: necrostatin-1; TNF-α: tumor necrosis factor-α.

**Figure 3 biomedicines-10-02913-f003:**
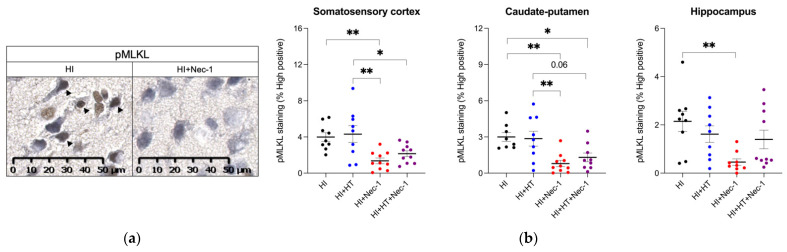
pMLKL expression in the brain at 5 h post-HI from pups exposed to HI±Nec-1±HT. (**a**) Representative image of pMLKL staining in the somatosensory cortex of HI compared to HI+HT+Nec-1-exposed animals. (**b**) Percentage of pMLKL staining compared to CTL within the somatosensory cortex, caudate-putamen, and hippocampus in HI±Nec-1±HT conditions. The number (n) of rats used was n = 7–8 per conditions from 8 litters. The bars indicate the mean ± SEM. * *p* ≤ 0.05; ** *p* ≤ 0.01, ANOVA with Tukey HSD’s multiple comparisons tests. Abbreviations: CTL: control; HI: hypoxia-ischemia, HT: hypothermia; Nec-1: necrostatin-1; pMLKL: phosphorylated forms of the mixed-lineage kinase domain-like protein.

**Figure 4 biomedicines-10-02913-f004:**
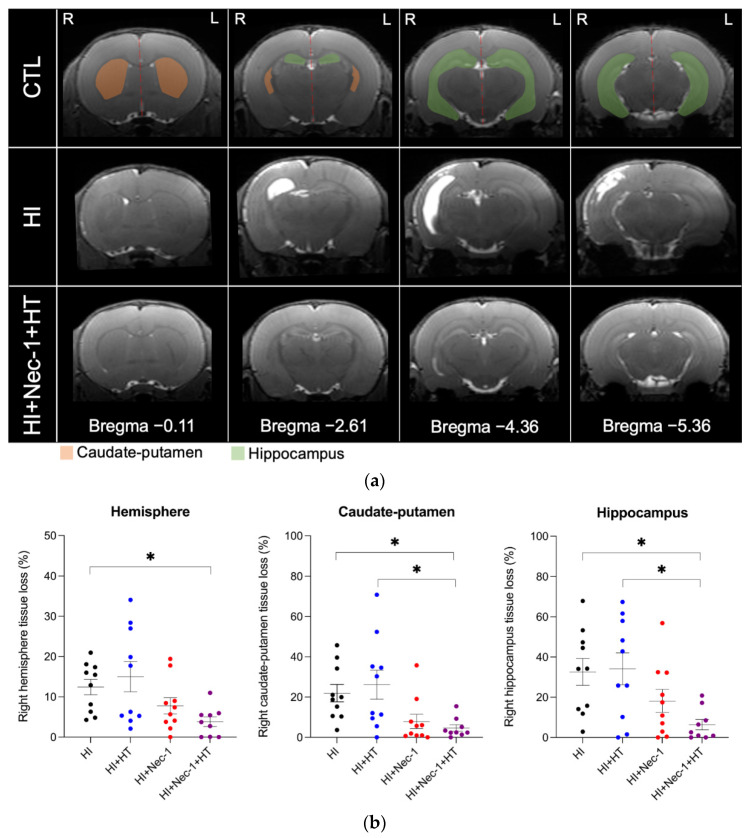
MRI brain volumes of HI±Nec-1±HT-exposed rats at P20-21. (**a**) Representative MRI images in CTL versus HI versus HI+Nec-1+HT conditions. (**b**) Ratio in percentages of tissue loss in the hemisphere, caudate-putamen, and hippocampus in HI±Nec-1±HT-exposed rats. The number (n) of rats used was n = 9–10 per conditions from 5 litters. The bars indicate the mean ± SEM. * *p* ≤ 0.05, Kruskal–Wallis with Dunn’s multiple comparisons tests. Abbreviations: CTL: control; HI: hypoxia-ischemia, HT: hypothermia; Nec-1: necrostatin-1.

## Data Availability

The datasets used and/or analyzed during the current study are available from the corresponding author on reasonable request.

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
