# Peer review of "Necroptosis Blockade Potentiates the Neuroprotective Effect of Hypothermia in Neonatal Hypoxic-Ischemic Encephalopathy"

_biomedicines, 2022, doi:10.3390/biomedicines10112913_

Round 1

Reviewer 1 Report

The study examined the role of inhibiting necroptosis in a rat model of hypoxic-ischemic encephalopathy.  The study found the necroptosis inhibitor Nec-1 had protective effects and these beneficial effects were enhanced with hypothermia treatment.  This is a well conducted study and manuscript is well written.

I have provided some suggestion to further improve manuscript.

Results

Section 3.2

Is it worth mentioning a positive trend for neuroprotection in the HI + Nec-1 treatment group; may be worth presenting P value as well (i.e., HI vs HI+ Nec-1).  I would predict that if Ns (e.g., to 15) were increased statistical significance would be reached

Discussion

Line 229: “In contrast to these studies, we did not observe Nec-1 effects on brain injury via MRI volume.”  I would argue on the contrary; this statement is too negative and needs to be measured in terms of the positive trend for neuroprotection with Nec-1 treatment (See comment above regarding Nec-1 data).

Other reasons why HT may not have been protective in this study is the delayed 30 min induction after HI, and the 4-hour duration of HT as opposed to immediate induction after HI and a 5-hour duration of HT (e.g., see ref 50).

Other

Really would have preferred to see a cohort of animals treated with Necrostatin-1 when administered after hypoxia.  Not having this cohort, reduces the potential clinical relevance of study as the usefulness of blocking the necroptosis pathway after HIE is not known.  I think this point needs to be discussed in the Discussion. 

Author Response

We thank the reviewer for his work on our manuscript. 

Reviewer 2 Report

31 October 2022

Regarding the review of manuscript Necroptotic blocking agent potentiation of hypothermia activity against neonatal hypoxic-ischemic encephalopathy’ by Chevin M et al., submitted to Biomedicines

Manuscript ID: biomedicines-1998711 

Dear Authors, 

Chevin and colleagues in the present research article entitled ‘Necroptotic blocking agent potentiation of hypothermia activity against neonatal hypoxic-ischemic encephalopathy’, investigated the beneficial effects of Necrostatin-1, a compound blocking necroptosis, in combination with hypothermia, in the care of neonatal hypoxia-ischemia. For this purpose, authors used a preclinical rat model of term human neonatal encephalopathy induced by hypothermia, and observed beneficial effects of Nec-1 against neonatal encephalopathy at the mechanistic (decreased of key neuroinflammatory markers) and anatomical (reduction in brain volume loss on MRI) levels.

The main strength of this manuscript is that it addresses an interesting and timely question, providing a captivating interpretation and describing mechanisms of cell death after acute neonatal brain injury, focusing on the beneficial effect of Nec-1 potentiation of hypothermia activity against neonatal hypoxic-ischemic encephalopathy (HIE).

In general, I think the idea of this article is really interesting and the authors’ fascinating observations on this timely topic may be of interest to the readers of Biomedicines. However, some comments, as well as some crucial evidence that should be included to support the author’s argumentation, needed to be addressed to improve the quality of the manuscript, its adequacy, and its readability prior to the publication in the present form. My overall opinion is to publish this paper after the authors have carefully considered my suggestions below, in particular reshaping parts of the ‘Introduction’ and ‘Methods’ sections by adding more evidence.

Please consider the following comments:

1.      Title: To aid the readers and maximize the accessibility to the manuscript, I believe that the title should be modified and should be read as a one concise sentence. Please, re-write the title ensuring that it is informative and appropriate

2.      Abstract: In my opinion, a lack of explanation and a brief description of what the ‘neonatal hypoxia-ischemia’ syndrome refers to could make the readers unable to grasp the key aspects of this article by consulting directly this section. Please, consider expanding this point and expand the abstract with 200 words, proportionally presenting the background (general, detailed, and the current issue addressed to this study), the objectives, the methods, the results, and the conclusion (the potential of this study and the advance this study has provided in this field).

3.      Keywords: I recommend expanding the keywords and using as many keywords as possible in the first two sentences of the abstract.

4.      A graphical abstract that will visually summarize the main findings of the manuscript is highly recommended.

5.      Introduction: The ‘Introduction’ section is well-written and nicely presented, with a good balance of descriptive text and information about the role of necroptosis in hypoxic-ischemic encephalopathy. Nevertheless, I believe that here the authors could focus on giving more specific information to better understand the precise mechanisms underlying necroptosis and its interactions with other cell death pathways in neurodegenerative diseases, as well as in other brain injuries, in order to provide significant therapeutic insights: in this regard, recent scientific literature has highlighted molecular mechanisms of necroptosis, the emerging evidence on necroptosis as a major driver of neuron cell death, and translational approaches to remedy (https://doi.org/10.3390/antiox11010039; https://doi.org/10.3390/cells11162607; https://doi.org/10.3390/ijms23136991; https://doi.org/10.3390/ijms23063149) Also, as necroptosis is involved in many pathological processes, such as trauma, cerebral ischemia-reperfusion injury, and inflammatory diseases, I believe that it could be useful adding results from recent studies that explored necroptosis in brains of Alzheimer’s disease patients, and how this is closely related to brain weight and cognitive dysfunctions (https://doi.org/10.1038/s41380-021-01326-4; https://doi.org/10.1111/psyp.14122).

6.      Measurement of the stroke volume: I suggest rewriting this section more accurately. Please provide more information about methods used to identify the borders of the lesions.

7.      In my opinion, the 'Results’ section is well organized, but it seems to state statistical significance of findings in an excessively broad way. Thus, I believe that this section would benefit from a more detailed and precise rewriting, in order to ensure in-depth understanding of the findings.

8.      Discussion: In this final section, authors described the results of their study and their argumentation and captured the state of the art well; however, I would have liked to see some views on a way forward. I believe that the authors should make their effort to explain the theoretical implication as well as the translational application of this paper, to adequately convey what they believe is the take-home message of their study and to present the ultimate goal, the knowledge and the technology necessary to achieve this goal, and future research directions.

9.      I would ask the authors to include a proper and defined ‘Limitations and future directions’ section before the end of the manuscript, in which the authors can describe in detail and report all the technical issues brought to the surface.

10.  Figures: I suggest modifying all figures for clarity and provide higher-quality images because, as it stands, the readers may have difficulty comprehending them. In my opinion, data settings are written with a very small font. In addition, all figures should be presented in color. 

Overall, the manuscript contains 5 figures, 1 table and 64 references. In my opinion, the manuscript might carry important value describing mechanisms of cell death after acute neonatal brain injury, focusing on the beneficial effect of Nec-1 potentiation of hypothermia activity against neonatal HIE. I hope that, after these careful revisions, this paper can meet the Journal’s high standards for publication.

I am available for a new round of revision of this article.

Best regards,

Reviewer

Author Response

Our responses to reviewer 1:

We thank this reviewer for acknowledging that “The main strength of this manuscript is that it addresses an interesting and timely question, providing a captivating interpretation and describing mechanisms of cell death after acute neonatal brain injury, focusing on the beneficial effect of Nec-1 potentiation of hypothermia activity against neonatal hypoxic-ischemic encephalopathy (HIE).

In general, I think the idea of this article is really interesting and the authors’ fascinating observations on this timely topic may be of interest to the readers of Biomedicines.” And that: “In my opinion, the manuscript might carry important value describing mechanisms of cell death after acute neonatal brain injury, focusing on the beneficial effect of Nec-1 potentiation of hypothermia activity against neonatal HIE.”

Regarding the title, in full compliance with the reviewer’s remark, we clarified our revised title as following: “Necroptosis blockade potentiates the neuroprotective effect of hypothermia in neonatal hypoxic-ischemic encephalopathy”.

  1. “Abstract: In my opinion, a lack of explanation and a brief description of what the ‘neonatal hypoxia-ischemia’ syndrome refers to could make the readers unable to grasp the key aspects of this article by consulting directly this section. Please, consider expanding this point and expand the abstract with 200 words, proportionally presenting the background (general, detailed, and the current issue addressed to this study), the objectives, the methods, the results, and the conclusion (the potential of this study and the advance this study has provided in this field).

In full agreement with Reviewer 1, we expanded the revised background section of our abstract (see page 1, lines 13-18).

  1. “Keywords: I recommend expanding the keywords and using as many keywords as possible in the first two sentences of the abstract.

Accordingly, we added keywords in the revised version of the manuscript.

  1. “A graphical abstract that will visually summarize the main findings of the manuscript is highly recommended.”

We suggest Figure 5 as a graphical abstract.

  1. Introduction: The ‘Introduction’ section is well-written and nicely presented, with a good balance of descriptive text and information about the role of necroptosis in hypoxic-ischemic encephalopathy. Nevertheless, I believe that here the authors could focus on giving more specific information to better understand the precise mechanisms underlying necroptosis and its interactions with other cell death pathways in neurodegenerative diseases, as well as in other brain injuries, in order to provide significant therapeutic insights: in this regard, recent scientific literature has highlighted molecular mechanisms of necroptosis, the emerging evidence on necroptosis as a major driver of neuron cell death, and translational approaches to remedy… Also, as necroptosis is involved in many pathological processes, such as trauma, cerebral ischemia-reperfusion injury, and inflammatory diseases, I believe that it could be useful adding results from recent studies that explored necroptosis in brains of Alzheimer’s disease patients...”

In agreement, we added some emerging evidence of the involvement of necroptosis in neurodegenerative diseases including amyotrophic lateral sclerosis (ALS), Alzheimer disease (AD), Parkinson disease (PD), and multiple sclerosis (MS) (please see page 1, lines 44-45; page 2, lines 62-67 and the added references 18 and 34).

  1. “Measurement of the stroke volume: “Please provide more information about methods used to identify the borders of the lesions.”

As suggested, we clarified this section by adding in our revised method section: “T2 hyperintensity was used as an effective marker to identify the boundaries of the lesion area on MRI, which were easily segmented using our image processing toolbox, as previously described (Chevin M, et al, 2020)”. See page 4, lines 161-163.

  1. In my opinion, the 'Results’ section is well organized, but it seems to state statistical significance of findings in an excessively broad way. Thus, I believe that this section would benefit from a more detailed and precise rewriting, in order to ensure in-depth understanding of the findings.

As advised, we carefully rechecked and validated the significance values in our results section. We specified in our revised results section the fold variation of some key results.

  1. “Discussion: In this final section, authors described the results of their study and their argumentation and captured the state of the art well; however, I would have liked to see some views on a way forward. I believe that the authors should make their effort to explain the theoretical implication as well as the translational application of this paper, to adequately convey what they believe is the take-home message of their study and to present the ultimate goal, the knowledge and the technology necessary to achieve this goal, and future research directions.

As suggested, we expanded our discussion to explain theoretical implication and translational application of Nec-1 as a potential therapy in a variety of human diseases, including stroke, ALS, AD, MS, and PD. Please see our revised discussion section (page 7, lines 282-291).

  1. I would ask the authors to include a proper and defined ‘Limitations and future directions’ section before the end of the manuscript, in which the authors can describe in detail and report all the technical issues brought to the surface.

We thank the reviewer for this point and added the above-mentioned section in our revised discussion section (page 7, line 273).

  1. “Figures: I suggest modifying all figures for clarity and provide higher-quality images because, as it stands, the readers may have difficulty comprehending them. In my opinion, data settings are written with a very small font. In addition, all figures should be presented in color.

All our figures have been designed in full compliance with the rules of Biomedicines (Cf. https://www.mdpi.com/journal/biomedicines/instructions). As requested, all possible figures were presented in color in our revised manuscript, except for anatomical magnetic resonance images (MRI), which are always black and white.

Reviewer 3 Report

My suggestions:

1. In the results section, I would add a table on the parameters of mouse groups (control, HT, HI) regarding the differences in inflammatory marker concentration

2. In the introduction I would discuss Nec-1 a little bit more in detail. Was it established to impact any kind of neurodegenerative disease?

3. In the discussion, instead of discussing Figure 5 in detail, I would add the explanation to the main text.

4. Is it possible that Nec-1 may protect against other neurodegenerative diseases, such as Alzheimer's disease, Parkinson's disease, etc?

5. Instead of a separate section for abbreviations, I would fully describe the abbreviation, when it was mentioned for the first time. 

Author Response

Our responses to reviewer 2:

“My suggestions:

  1. In the results section, I would add a table on the parameters of mouse groups (control, HT, HI) regarding the differences in inflammatory marker concentration”

Respectfully, we would mention to reviewer 2 that we did not make any measure of concentration of inflammatory markers in this study. We only used semi-quantitative IHC staining.

  1. “In the introduction I would discuss Nec-1 a little bit more in detail. Was it established to impact any kind of neurodegenerative disease?”

As suggested, we discussed more in detail the role of necroptosis in other neurodegenerative diseases (please, see our revised Introduction section, page 2, lines 62-67).

  1. “In the discussion, instead of discussing Figure 5 in detail, I would add the explanation to the main text.”

Accordingly, we transferred a great part of our initial Figure 5 legend in the main text of the revised version of our manuscript (see page 7, lines 279-291).

  1. “Is it possible that Nec-1 may protect against other neurodegenerative diseases, such as Alzheimer's disease, Parkinson's disease, etc?”

As requested, we further discussed the potential of Nec-1 as therapeutics in a variety of neurodegenerative diseases, including ALS, MS, AD, and PD (see our revised discussion section, page 7, lines 282-291).

  1. Instead of a separate section for abbreviations, I would fully describe the abbreviation, when it was mentioned for the first time.

As requested, we fully describe abbreviations when mentioned for the first time.

Reviewer 4 Report

The paper is well written, except few minor points reported below. It is clear to follow and understand.

Material & Methods. Results

Table 1 would be better as supplementary table

Did the authors perfomed RNA studies? Gene expression data would be helpful to correlate with immunohistochemistry results.

It would be good to add experimental data about expression of TNF-alfa, RIPK3 and MLKL genes. 

Author Response

Our responses to reviewer 3:

We thank the reviewer for stating that “The paper is well written, except few minor points reported below. It is clear to follow and understand”.

  1. Table 1 would be better as supplementary table.

As requested, we put the Table 1 as a supplementary figure.

  1. Did the authors performed RNA studies? Gene expression data would be helpful to correlate with immunohistochemistry results. It would be good to add experimental data about expression of TNF, RIPK3 and MLKL genes.

We thank reviewer for this interesting point. However, we think that such study is beyond the scope of this manuscript. We will consider it in our next step of research. We added as a limitation in our revised discussion section that: “our study involved in-situ IHC methods but mRNA explorations were not carried out. Future studies should ideally consider testing for gene expressions involved in necroptosis-induced brain injuries in HIE” (please see our revised limitations section, page 7, lines 279-282).

Round 2

Reviewer 2 Report

7 November 2022 

Regarding the 2nd review of manuscript Necroptotic blocking agent potentiation of hypothermia activity against neonatal hypoxic-ischemic encephalopathy’ by Chevin M et al., submitted to Biomedicines

Manuscript ID: biomedicines-1998711 

Dear Authors, 

In this article Chevin and colleagues investigated the beneficial effects of necrostatin-1, a compound blocking necroptosis, in combination with hypothermia, in the care of neonatal hypoxia-ischemia. I am pleased to see that the authors took my comments seriously and solved many issues I have raised in the previous round of the review session.

I only have few last suggestions to do further to improve the theoretical background of the present paper and its argumentation, and thus, I believe, to improve the quality of the manuscript, before I finalized my part of the peer-review session.

Comments:

1.      Abstract: Please abridge the abstract to 200 words according to the journal’s guidelines. The background is well written, but it has become too long, compared to other subsections, while there is no background on the relationship between necroptosis and neonatal hypoxic-ischemic encephalopathy including necrostatin. The authors need to add more description of the methods. Overall, it is important to pay attention to the proportion of the background, the objectives, the methods, the results, and the conclusion.  

2.      Introduction: I believe that the manuscript would benefit from more specific information on the precise mechanisms underlying necroptosis and its interactions with other cell death pathways in neurodegenerative diseases, as well as in other brain injuries (https://doi.org/10.3390/cells11162607; https://doi.org/10.3390/ijms23136991) and on how this is closely related to brain weight and cognitive dysfunctions (https://doi.org/10.1038/s41380-021-01326-4; https://doi.org/10.1111/psyp.14122).

Overall, this is a timely and needed study, and I look forward to seeing further studies on this issue by these authors in the future. I am always available for other revisions of such as interesting and important studies.

Thank You for your work.

Best regards,

Reviewer

Author Response

Dear Editor,

We would like to thank the reviewers for their remarks, which improved our manuscript.

Please find hereunder, our point-by-point answers to these remarks.

  1. Abstract: Please abridge the abstract to 200 words according to the journal’s guidelines. The background is well written, but it has become too long, compared to other subsections, while there is no background on the relationship between necroptosis and neonatal hypoxic-ischemic encephalopathy including necrostatin. The authors need to add more description of the methods. Overall, it is important to pay attention to the proportion of the background, the objectives, the methods, the results, and the conclusion.  

In full agreement with the Reviewer, the abstract has been abridged to 200 words. As requested, the background has been reduced and we pay attention to the proportion of the other sections.

  1. Introduction: I believe that the manuscript would benefit from more specific information on the precise mechanisms underlying necroptosis and its interactions with other cell death pathways in neurodegenerative diseases, as well as in other brain injuries (https://doi.org/10.3390/cells11162607; https://doi.org/10.3390/ijms23136991) and on how this is closely related to brain weight and cognitive dysfunctions (https://doi.org/10.1038/s41380-021-01326-4; https://doi.org/10.1111/psyp.14122).

Accordingly, we believe that this has been largely described in the Introduction and Discussion sections of our revised manuscript (please see page 2 lines 62-67, and page 7 lines 299-304, as well as Figure 5) Respectfully, we do not find the link in between the references mentioned by the Reviewer and necroptosis. As per journal guidelines, we make sure that all references are relevant to the content of the manuscript. Therefore, please see the references 34, 57 - 61 in our revised manuscript, which were used to describe mechanisms underlying necroptosis and its interactions with other cell death pathways in neurodegenerative diseases and other brain injuries.

We thank the Reviewer for his important work during the peer-review process and believe his suggestions were fully taken into the account and improved the quality of our manuscript.

Sincerely,

Mathilde Chevin, PhD.

Reviewer 3 Report

The authors fulfilled my suggestions

Author Response

We thank the Reviewer for his work on our manuscript.

Sincerely,

Mathilde Chevin, PhD.